# The Analysis of Causality and Risk Spillover between Crude Oil and China’s Agricultural Futures

**DOI:** 10.3390/ijerph191710593

**Published:** 2022-08-25

**Authors:** Wei Jiang, Ruijie Gao, Chao Lu

**Affiliations:** School of Economics, Hangzhou Normal University, Hangzhou 311121, China

**Keywords:** green agricultural transformation, sustainable agricultural development, time-varying Granger causality test, DY dynamic spillover index

## Abstract

This paper aims to apply the time-varying Granger causality test (TVGC) and the DY Spillover Index (Diebold and Yilmaz, 2012) to measure the Granger causality and dynamic risk spillover effects of the international crude oil futures market on China’s agricultural commodity futures market from the perspectives of return and volatility spillovers. Empirical evidence relating to the TVGC test suggests the existence of unidirectional Granger causality between crude oil futures and agricultural product futures. This relationship shows a strong time-varying property, in particular for sudden or extreme events such as financial crises and natural disasters. On the other hand, the volatility spillover in crude oil and agricultural product futures markets responds asymmetrically and bidirectionally according to the result of the DY Spillover index, and the periodicity of total volatility spillover correlates closely with the occurrence of global economic events, which indicates that the spillover effect between crude oil and agricultural commodity futures markets will be exacerbated in turbulent financial and economic times. Such findings are expected to help in formulating policy recommendations, portfolio design, and risk-management decisions.

## 1. Introduction

The scope of China’s participation in global trade has expanded rapidly in recent years. A competitive environment has gradually generated in the agricultural product market due to the close link between domestic and international markets. As crude oil is a primary source of energy for agricultural production, its price changes will inevitably cause variations in agricultural commodities on international markets. In terms of the futures market, a large number of investors and speculators have been attracted to participate in crude oil futures trading ever since such derivatives were launched by the futures exchange. These active transactions may bring about great uncertainties and dramatic price swings not only to the crude oil market but also to other commodity markets. In addition, with the spread of the global food crisis, the agricultural market will, in turn, affect other commodity markets. As the world’s largest importer of crude oil and agricultural products, it is essential to figure out the causal relationship and dynamic interaction characteristics between the crude oil futures market and China’s agricultural product futures market. The outcomes of the study are expected to provide reference information for avoiding a global systemic food crisis.

Theoretically, international crude oil prices can affect China’s agricultural product prices in three major ways: (1) agricultural trade. Associated with international agricultural product prices, China’s agricultural products usually respond consistently to international standards. It is well-known that the rise of international crude oil can cause international agricultural product prices to change in the same direction. Such a shift drives up the domestic price levels due to higher import costs. (2) The production cost of agricultural products. As a net importer of crude oil, China’s oil price will rise when the international crude oil price goes up. Higher oil price brings extra costs to production, processing, and transportation. Correspondingly, suppliers will increase the price of agricultural products in order to keep their own profits. (3) Biofuel development. The Chinese government attaches primary importance to the growing climate problem and has encouraged the development of a green economy over the years. Biofuels produced from agricultural raw materials are gradually replacing oil and being supplied in various fields with their own advantages, such as recyclability and environmental protection. Once the crude oil price rises substantially, China will reduce the demand for oil and turn to biofuels instead. As a result, the increased consumption of raw materials such as soybean meal and corn will promote agricultural prices to rise, especially in cases when demand exceeds supply.

It is therefore critical to find out the impact of the crude oil price changes on agricultural product markets. The objective of this paper is to explore the internal mechanism of volatility in the agricultural futures market and crude oil futures market so that the latter can be monitored to improve the early warning mechanism for oil prices. The major contributions are as follows: First, our research takes both the Granger causality of returns and the spillover effect of volatilities into consideration, while most previous studies focus only on the relationship of different return series. Second, the sample size covers almost 20 years and reveals the dynamics of causality and spillover during stable and volatile periods, so that risk managers and investors can predict returns and volatilities of one market based on the other market. Third, we find that the current decrease in the net spillover of crude oil futures to agricultural futures is mainly attributed to the promotion of biofuel energy. These findings will play a supportive role in enabling the macro-department of the Chinese government to formulate relevant policies that contribute to controlling risks and stabilizing the agricultural product market.

The remainder of this paper is organized as follows. Section 2 presents the relevant previous literature. Section 3 introduces the methodology. Data and empirical results are analyzed in Section 4 and Section 5, respectively. Section 6 concludes the paper.

## 2. Literature Review

Previous works use different models for various financial assets and time scales to investigate the futures price relationship between crude oil and agricultural products. With the continuous improvement in econometrical methods, research on the aspects of the Granger causality test and risk-spillover effects has become increasingly in-depth.

It has been reported in many empirical studies that the international crude oil price has a significant impact on the prices of agricultural products. Further research even found mutual influence between the two. Specifically, by using the panel VAR and the Granger causality test to examine the relationship between crude oil prices and agricultural prices, a study documented a bidirectional causal relationship between the two [1]. Meanwhile, such a relationship was also pointed out by some scholars, where the long-term correlation and causal relationship were tested by Granger causality analysis [2]. By applying the ARDL method combined with Granger causality to examine the dynamic relationship among crude oil, biofuels, and agricultural prices, the results showed that there is a strong dependence in both long and short terms [3]. By adopting the panel method and cointegration test to analyze the dynamic relationship between crude oil and agricultural products during 2006–2015, the study showed that the rise of oil prices could correspondingly push agricultural prices higher [4]. Linear and Non-linear Autoregressive Distributed Lag (ARDL) Models are often used by scholars to investigate the impact of oil price shocks on agricultural prices, and a large number of research results suggest that agricultural products and crude oil are co-moving in the long run [5,6,7]. By applying the Panel-VAR model to energy prices and food prices during the years 2000–2016, the results of the impulse response function demonstrated a positive response to any shock to oil prices [8]. Some references considered using the time-varying rolling window technique to explore the causal relationship in prices between oil and agricultural products. The research showed the existence of a time-varying positive bidirectional causal relationship in a certain period of time [9].

On the other hand, some scholars hold the opposite view that there is no correlation with respect to prices between international crude oil and agricultural products. The Copula model was employed to study the co-movement of prices in crude oil and several typical agricultural products (corn, soybeans, and wheat), where no extreme market dependence was found between crude oil and agricultural product prices, indicating that the impact of agricultural product markets on crude oil was neutral [10]. A SVAR, along with a direct cyclic graph, was employed to decompose how supply/demand structural shocks affect food and fuel markets. Empirical results supported the hypothesis that fundamental market forces of demand and supply are the main drivers of food price volatility, while the shocks from oil, gasoline, and ethanol markets did not spill over into grain prices in the long run [11]. The study argued that the price of agricultural products in South Africa is neutral to the fluctuation of oil price based on the results of the structural mutation cointegration test and nonlinear causality test [12].

The existence of spillover effects will help investors, risk managers, manufacturers, and policymakers to capture the demand for commodity futures prices dynamically [13,14]. Since the outbreak of the 2008 global financial crisis, various sorts of data and econometric models have been selected to investigate the spillover effects between crude oil and agricultural products. For instance, by using the causal variance test and impulse response function to examine the volatility transmission between oil and agricultural prices from 1986 to 2011, research found that despite there being no volatility spillover before the financial crisis, oil volatility transmitted to agricultural products was detected after the financial crisis [15]. Three different GARCH models were employed to catch the correlation between crude oil and energy crops by some scholars. Results from such dynamic models exhibit a strong correlation of about 20 percent in regard to daily returns. Furthermore, they further used the frequency-dependent spillovers measure to explore return spillovers from crude oil to ethanol, corn, soybean, and wheat and showed return spillover is stronger only during periods of energy and food crisis [16,17]. A multifractal detrended cross-correlation analysis approach was utilized to analyze the cross-correlations between the Brent crude oil and agricultural futures. The experimental results indicated that the multifractal cross-correlation was stronger under the influence of the COVID-19 pandemic [18]. A relational measurement based on Markov-switching GRG copula was constructed to analyze the dependence structure between futures prices of WTI crude oil and 12 kinds of Chinese agricultural commodities. The degree of correlation with crude oil futures prices varies under different agricultural commodity futures prices [19]. Some scholars examined the nature and dynamics of volatility spillovers during the period of the 2008–2009 financial crisis via the bivariate heterogeneous autoregressive model, from which bidirectional spillovers were observed between crude oil and agricultural commodity markets [20].

Along with the improvement in related models and methods, Diebold and Yilmaz proposed a DY spillover index in 2009 to measure the spillover effect of return and volatility spillovers. Such an index was based on the forecast-error variance decompositions and was improved later by them with a generalized variance decompositions framework to avoid the sequence-dependence problem in 2012. By using this approach in the US stock, bond, foreign exchange, and commodity markets, they concluded that with the deepening of the financial crisis, volatility spillover effects also increased subsequently. Moreover, they proposed several connectedness measures and focused on the average and daily time-varying connectedness of major US financial institutions’ stock return volatilities in recent years, including during the financial crisis of 2007–2008 [21,22,23]. Some scholars used the spillover index method to describe the relationship between the volatility of corn and energy prices in 2018 [24]. By combining a multivariate heteroscedastic autoregressive (HAR) model with the DCC-GARCH model to analyze the connectedness characteristics between US crude oil futures and China’s agricultural commodity futures, the results verified the existence of leverage volatility transmission across markets [25].

Spillover effects include the mean spillover effect and the volatility spillover effect. The former refers to the impact of a specific commodity price change on the price level of other commodities, while the latter represents the impact of volatility for one certain commodity on other commodities [26]. A number of studies on the spillover effect between crude oil and agricultural commodities started with these two perspectives. Against this background, a fractionally integrated VAR model was employed to capture the long-memory behavior of the implied volatilities alongside the Markov Switching Autoregressive model to extract the regimes of crude oil. Evidence showed that the net volatility spillover effect from crude oil to all agricultural commodities tends to decrease when crude oil remains in its low-volatility regime. Conversely, this effect experienced an increasing trend when crude oil remained in its relatively high-volatility regime [27]. An analysis of the spillover effect and time-frequency connectedness between crude oil prices and agricultural commodity markets was conducted by some scholars. Via the DY spillover index and the wavelet coherence model, a more apparent mean spillover was revealed during the COVID-19 pandemic [28]. Some references examined spillover effects by employing the DY spillover index to returns and volatilities. The findings indicated an asymmetric and bidirectional flow of information among crude oil and agricultural commodities that intensifies during periods of financial and economic turmoil [29].

It is important to research the relationship between energy and agricultural commodity markets, especially for investors’ portfolio optimization, risk management, and asset allocation. Despite the fact that many existing studies have explored such an issue, their conclusions are not consistent with each other. Methodologically, VAR, MGARCH, and Copula are frequently used in the current literature to analyze volatility spillover. These models, however, failed to provide information with respect to the direction of volatility spillover, which therefore may lead to opaque dynamic spillover effects. To cope with these problems, this paper applies the time-varying Granger model and the DY spillover index model to examine the dynamic Granger causality that exists in crude oil and agricultural futures and identify the direction of volatility spillover of the investigated markets.

## 3. Methodology

### 3.1. Time-Varying Granger Causality Tests

The fundamental idea of the Granger causality test is to determine whether one sequence is useful in terms of forecasting another sequence. That is, if the prior values have an explanatory ability to predict the future values of another time series, there should be a causal link between the two variables. The Granger causality test is directly related by sample period so that data in different time spans may result in different conclusions; we therefore alternatively use the time-varying Granger causality test to test the causal relationship between crude oil and agricultural futures markets.

A brief introduction to the Granger causality framework is as follows. We can consider a VAR(*m*) model including two variables:(1)$$y_{1t}=\phi_{0}^{\left(1\right)}+\overset{m}{\underset{k=1}{\sum }}\phi_{1k}^{\left(1\right)}y_{1,t-k}+\overset{m}{\underset{k=1}{\sum }}\phi_{2k}^{\left(1\right)}y_{2,t-k}+\varepsilon_{1t}$$
(2)$$y_{2t}=\phi_{0}^{\left(2\right)}+\overset{m}{\underset{k=1}{\sum }}\phi_{1k}^{\left(2\right)}y_{1,t-k}+\overset{m}{\underset{k=1}{\sum }}\phi_{2k}^{\left(2\right)}y_{2,t-k}+\varepsilon_{2t}$$
where $y_{1t}$ and $y_{2t}$ stand for two different time series. Variable $y_{1}$ is referred to as the Granger cause of a variable $y_{2}$ if the current value of $y_{2}$ can be predicted by the historical values of $y_{1}$. The Wald test is used for testing the joint significance of parameters $\phi_{1k}^{\left(2\right)}\;\left(k=1,\cdots ,m\right)$, whose null hypothesis is no Granger causality between $y_{1}$ and $y_{2}$. The matrix structure for VAR(m) can be expressed as
(3)$$y_{t}=\Pi x_{t}+\varepsilon_{t}$$
where $y_{t}={\left(y_{1t},\;y_{2t}\right)}^{\prime }$, $x_{t}=\left(1,y_{t-1}^{\prime },\;y_{t-2}^{\prime },\cdots ,y_{t-k}^{\prime }\right){}^{\prime }$, $\Pi_{2\times \left(2m+1\right)}=\left(\phi_{0},\phi_{1},\cdots ,\phi_{m}\right)$, and $\phi_{0}={\left(\phi_{0}^{\left(1\right)},\phi_{0}^{\left(2\right)}\right)}^{\prime }$, $\phi_{k}=\left(\begin{array}{cc} \phi_{1k}^{\left(1\right)} & \phi_{2k}^{\left(1\right)}\\ \phi_{1k}^{\left(2\right)} & \phi_{2k}^{\left(2\right)} \end{array}\right)$, k=1,2,⋯,m. The null hypothesis of the Granger causality test for variables $y_{1}$ and $y_{2}$ is $R_{1\to 2}\pi =0$, where $R_{1\to 2}$ is the coefficient restriction matrix, and π is the row-vectorized vec(Π).

The Wald statistic modified for heteroscedasticity $w_{1\to 2}$ is defined as
(4)$$w_{1\to 2}=T{\left(R_{1\to 2}\hat{\pi }\right)}^{\prime }{\left[R_{1\to 2}\left({\hat{V}}^{-1}\hat{\Sigma }{\hat{V}}^{-1}\right)R_{1\to 2}^{\prime }\right]}^{-1}\left(R_{1\to 2}\hat{\pi }\right)$$
where $\hat{V}=I_{n}⊗\hat{Q}$, $\hat{Q}=T^{-1}\Sigma_{t}x_{t}x_{t}^{\prime }$, $\hat{\Sigma }=T^{-1}\Sigma_{t}\hat{\xi_{t}}\hat{\xi_{t}^{\prime }}$, and $\hat{\xi_{t}}=\hat{\varepsilon_{t}}⊗x_{t}$, $\hat{\varepsilon_{t}}=y_{t}-\hat{\Pi }x_{t}$.

### 3.2. Spillover Index Frameworks

Diebold and Yilmaz (2009) proposed the DY spillover index, which is based on the decomposition of the forecast-error variance of the VAR model, to measure the spillover effect for different variables. Such an approach was improved later in Diebold and Yilmaz (2012) using a generalized variance decompositions framework in which forecast-error variance decompositions are invariant to the variable ordering. This paper implements the advanced index for measuring the spillover effect over crude oil and agricultural futures.

The VAR(p) model, including N variables, takes the form of
(5)$$x_{t}=\overset{p}{\underset{i=1}{\sum }}\varphi_{i}x_{t-i}+\varepsilon_{t}$$

Random variables are assumed to be independent and identically distributed, i.e., $\varepsilon_{t}\sim i.i.d.N\left(0,\Sigma \right)$. Through a moving average, Equation (5) can be rewritten as given below.
(6)$$x_{t}=\overset{\infty }{\underset{i=0}{\sum }}A_{i}\varepsilon_{t-i}$$
where $A_{i}$ denotes the N×N identity matrix and satisfies the following expression.
(7)$$A_{i}=\varphi_{1}A_{i-1}+\varphi_{2}A_{i-2}+\cdots +\varphi_{p}A_{i-p}$$

Particularly, $A_{i}=0$ if i<0. This moving average coefficient is the key to the VAR model. Variance decomposition attributes the forecast-error variance decomposition of the respective variables to the shocks of other variables within a specific system. Variance decomposition requires orthogonalized information. The information in the VAR model, however, is contemporaneously correlated. Although Cholesky decomposition can realize orthogonalization, the result of variance decomposition depends on variable ordering. For this purpose, a generalized VAR model is constructed to cope with the ordering problem. The DY dynamic spillover index is based on the generalized variance decomposition. By calculating the variance component, it is possible to obtain the total spillover, the directional spillover, the net spillover, and the net pairwise spillover, respectively.

The variance component is defined as the score of the H-period forecast-error variance with respect to the shock of variable $x_{i}$ to itself, while the covariance component or spillover represents the score of the H-period forecast-error variance with respect to the shock of variable $x_{i}$ to variable $x_{j}$, where i,j=1,2,⋯,N and i≠j.

Let $\theta_{ij}^{g}\left(H\right)$ be the H-period forecast-error variance decomposition obtained from the KPPS method under the generalized VAR framework. For H=1,2,⋯, $\theta_{ij}^{g}\left(H\right)$ is calculated by
(8)$$\theta_{ij}^{g}\left(H\right)=\frac{\sigma_{jj}^{-1}\sum_{h=0}^{H-1}{\left(e_{i}^{\prime }A_{h}\Sigma e_{j}\right)}^{2}}{\sum_{h=0}^{H-1}\left(e_{i}^{\prime }A_{h}\Sigma A_{h}^{\prime }e_{i}\right)}$$
where Σ is the variance–covariance matrix of the error vector ε, $\sigma_{jj}$ is the standard deviation of the error term $\varepsilon_{j}$, and $e_{i}$ is a vector whose i-th element is 1 and the remaining elements are 0. Then, normalizing the variance decomposition matrix by row so that they sum to unity:(9)$${\hat{\theta }}_{ij}^{g}\left(H\right)=\frac{\theta_{ij}^{g}\left(H\right)}{\sum_{j=1}^{N}\theta_{ij}^{g}\left(H\right)}$$

Accordingly, $\sum_{j=1}^{N}{\hat{\theta }}_{ij}^{g}\left(H\right)=1$ and $\sum_{i,j=1}^{N}{\hat{\theta }}_{ij}^{g}\left(H\right)=N$.

Therefore, the total volatility spillover index used to measure the contribution of spillovers from all market shocks to the total forecast-error variance in a generalized VAR can be constructed as follows, given the variance contribution rate computed by KPPS variance decomposition.
(10)$$S^{g}\left(H\right)=\frac{\sum_{i,j=1,i\neq j}^{N}{\hat{\theta }}_{ij}^{g}\left(H\right)}{\sum_{i,j=1}^{N}{\hat{\theta }}_{ij}^{g}\left(H\right)}\times 100=\frac{\sum_{i,j=1,i\neq j}^{N}{\hat{\theta }}_{ij}^{g}\left(H\right)}{N}\times 100$$

In view that the generalized impulse response and variance decomposition are invariant to variable ordering, we use the elements of the normalized generalized variance decomposition matrix to calculate the directional spillover effect. In a measurement system, Equation (11) describes the directional spillover received by variable i from all other variables j.
(11)$$S_{i\leftarrow j}^{g}\left(H\right)=\frac{\sum_{j=1,j\neq i}^{N}{\hat{\theta }}_{ij}^{g}\left(H\right)}{\sum_{i,j=1}^{N}{\hat{\theta }}_{ij}^{g}\left(H\right)}\times 100$$

Similarly, the directional spillover transmitted from variable i to all other variables j can be expressed as
(12)$$S_{i\to j}^{g}\left(H\right)=\frac{\sum_{j=1,j\neq i}^{N}{\hat{\theta }}_{ji}^{g}\left(H\right)}{\sum_{i,j=1}^{N}{\hat{\theta }}_{ji}^{g}\left(H\right)}\times 100$$

Thus, a set of directional spillovers can be recognized as spillovers with specific sources decomposed from the total spillover.

In addition, the net volatility spillover effect from variable i to the other j variables identifies whether i is a source or recipient of spillovers.
(13)$$S_{i}^{g}\left(H\right)=S_{i\to j}^{g}\left(H\right)-S_{i\leftarrow j}^{g}\left(H\right)$$

The net volatility spillover provides aggregated information with respect to the net contribution of a specific variable to the other variables. The net pairwise volatility spillover effect between variables i and j measures the difference between the total volatility shock transmitted from variables i to j and variables j to i.
(14)$$S_{ij}^{g}\left(H\right)=\left[\frac{{\hat{\theta }}_{ji}^{g}\left(H\right)}{\sum_{i,k=1}^{N}{\hat{\theta }}_{ik}^{g}\left(H\right)}-\frac{{\hat{\theta }}_{ij}^{g}\left(H\right)}{\sum_{j,k=1}^{N}{\hat{\theta }}_{jk}^{g}\left(H\right)}\right]\times 100$$

## 4. Data

### 4.1. Variable Selection and Data Sources

This section applies the returns and volatilities of the WTI crude oil futures, the agricultural futures index, and three representative Chinese agricultural futures consisting of wheat hard futures, cotton futures, and soybean meal futures for empirical analysis. The datasets provided by NYMEX, iFinD, and RESSET are during the periods of June 2004–May 2022, and the sample sizes are all 2825. It is noticeable that this paper only focuses on the overlapping parts of trading time to guarantee the availability of data, and the missing values are filled by the linear-interpolation method.

Suppose that the close prices of specific futures at day t−1 and day t are $P_{t-1}$ and $P_{t}$, respectively. Based on daily transaction prices data, the return of day t without considering the dividend can be calculated by $r_{t}=\ln\left(P_{t}/P_{t-1}\right)\times 100$. Let WTI, AFI, WHF, CF, and SMF be the logarithm of the return series of the WTI crude oil futures, the agricultural futures index, the wheat hard futures, the cotton futures, and the soybean meal futures, respectively. The counterpart volatilities denoted as WTI-σ, AFI-σ, WHF-σ, CF-σ, and SMF-σ are obtained by fitting the ARMA(1,1)-GARCH(1,1)-t distribution model.

### 4.2. Descriptive Statistics

Table 1 summarizes the descriptive statistics of log daily returns and log volatilities for the respective futures, where JB is the *p*-value of Jarque–Bera statistics. The standard deviation results suggest that the largest fluctuations appeared in the cases of SMF, WTI, SMF-σ, and WTI-σ, followed by CF and WHF. The fluctuation of AFI is relatively low. Despite the fact that the skewness of CF is not significantly different from zero, such a measure shifts to left or right in the other returns data, while all the volatilities are right-skewed. The kurtosis indicates the return distributions are leptokurtic, as is commonly observed in financial returns. The skewness and kurtosis imply that the investigated data are not distributed in a Gaussian fashion, and the Jarque–Bera statistics reject the null hypothesis of normality, even at the 1% significance level with respect to all variables. Furthermore, the null hypothesis of the Augmented Dicky–Fuller (ADF) unit root test is rejected at the 1% significance level, except for WHF-σ. Nevertheless, all series perform stationary at the 10% significance level.

## 5. Empirical Analysis

### 5.1. Time-Varying Granger Causality between Return Series

#### 5.1.1. Lag Order and Stability Discrimination for VAR Model

Determining a proper lag order is essential to the establishment of the VAR model. In general, the selection standards usually follow the LR, FPE, AIC, SC, and HQ information criteria. Table 2, Table 3, Table 4 and Table 5 calculate that the lag orders for AFI, WHF, CF, and SMF should be 1, 3, 3, and 4, respectively. Moreover, as plotted in Figure 1, all eigenvalues of the established models are within the unit circle, which indicates these VAR models are stable.

#### 5.1.2. Granger Causality Test

The Granger causality test on the return series is implemented in advance to clarify the transmission direction between the returns of international crude oil and China’s agricultural futures. Table 6 shows that WTI is the Granger cause of AFI, WHF, CF, and SMF. In the meantime, SMF is also the Granger cause of WTI, while the other three futures are not. The existence of the bidirectional Granger causality between WTI and SMF means these two futures are mutually influenced by each other. In contrast, price changes in AFI, WHF, and CF are not supposed to affect WTI.

#### 5.1.3. Time-Varying Granger Causality Test

The Granger causality test relies directly on time periods so that different time series samples may lead to different results. Therefore, this paper further implements the time-varying Granger causality test to verify whether crude oil and agricultural futures returns have time-varying Granger causality.

The window scale is set to 250, and the lag orders with regard to the respective sequences are set to be equal to those obtained in the conventional Granger causality test. By assuming the error terms to be identically distributed, this paper performs three types of window forms, including forward, rolling, and recursive ones, to test the Granger causality in agricultural and crude oil markets. In terms of specification, the forward expanding (FE) window test first calculates the Wald statistics with the minimum window size, then expands the window length by one observation in succession until the entire sample is used to compute the statistics. The rolling (RO) window test also moves forward by one observation at a time, whereas each window is rolled with a fixed sample size and acquires one Wald statistic. As an extension to the aforementioned approaches, the recursive evolving (RE) window test relies on repeated estimation on a forward-expanding sample sequence and is only restricted by the minimum window size.

Table 7 gives the Wald test results for time-varying Granger causality. The null hypothesis is rejected at the 10% significance level for WTI→SMF and at the 5% significance level for the others, demonstrating that correlations between crude oil and agricultural commodities exist over time. Moreover, test results against WTI→AFI are particularly depicted in Figure 2 for a more detailed analysis with respect to the changes of the Granger causality between crude oil and agricultural futures. Figure 2a–c exhibit the forward, rolling, and recursion procedures under the condition that error terms are homoskedastic, while the corresponding Figure 2d–f take the heteroskedasticity into account. As shown in Figure 2, the unidirectional Granger causality test for *WTI* → *AFI* is time-varying.

Specifically, Figure 2a–c plot the Wald statistics sequences based on three different windows, from which the time-varying property can be obviously captured. In the rolling algorithm, WTI is the Granger cause of AFI during three separate time periods of 2008–2009, 2012–2013, and 2015–2018, while in the forward and recursive algorithms, the time periods are 2008–2022 and 2007–2022, respectively.

Counting the major world events along the respective timeline, we found that Hurricane Katrina hit the Gulf of Mexico and caused tremendous damage in August 2005. The subprime mortgage crisis broke out in 2008 and led to stock markets plunging around the world in the following years. The European debt crisis continued during the years 2008–2011. Additionally, recent years have witnessed the widespread economic shocks brought by the COVID-19 pandemic since the first case was reported at the end of the year 2019. From the time nodes, we can conclude that the prices of the agricultural futures index fluctuate only when the prices of crude oil are affected by sudden financial events or natural disasters. That is, crude oil is the Granger cause of the agricultural futures index. This dynamic analysis suggests that the relationship between WTI and AFI is sensitive to the sample period. Only under certain conditions, the WTI crude oil could be the Granger-cause of the agricultural futures index. In addition, Figure 2d–f present the Wald test results considering heteroskedasticity as a robust test for the homoskedasticity cases, in which virtually all time periods that show significant results for the Granger causality test are consistent with Figure 2a–c.

### 5.2. Empirical Analysis of Volatility Spillover

This paper applies the volatility spillover index model to estimate the information transfer effect along with the dynamic process across different futures markets between crude oil and agricultural commodities.

#### 5.2.1. Total Spillover Index

The spillover effect explains the variance component of different futures in the market. Information given by a spillover index matrix reveals the varying contribution rates of the forecast-error variance decomposition with respect to all variables, in which the diagonal and non-diagonal elements reflect the spillover effects caused by internal changes and cross-market changes, respectively.

Table 8 illustrates the total static volatility spillovers based on vector autoregressions of order two (determined by the AIC and FPE criteria) and generalized variance decompositions of 10-day-ahead forecast errors, where “From” and “To” compute the degrees of spillovers received from and transmitted to other markets. “NS” represents the net spillovers that measure the difference between “To” and “From”, and a negative value indicates a net spillover receiver, while a positive value means a net spillover transmitter. The total spillover displayed in the lower-right corner of the table is estimated to be 7.57, showing that, on average, 7.57% of the forecast-error variance is attributed to other markets.

According to the estimation results, several conclusions can be drawn: (1) The own-variable spillovers are generally higher than the cross-variable spillovers with respect to both crude oil and agricultural futures markets, especially for the case of WTI-σ, where 97% of the spillover is caused by its own volatility. (2) Regarding the bidirectional volatility spillover effects, AFI-σ is the largest transmitter and receiver to the other futures with respective contribution rates of 12.99% and 13.75% on average. In contrast, volatility spillovers from crude oil to agricultural futures is 2.80%, and the reverse transmission is 3.00%, demonstrating the fact that crude oil is a net volatility-spillover-receiver from agricultural futures markets. (3) The interconnection among the agricultural sector plays an important role in affecting volatilities. For example, the largest net transmitter of volatility spillovers to the other agricultural futures is SMF-σ, whose net contribution is 1.02%, followed by CF-σ, with a net spillover of 0.89%, while the remaining agricultural futures are net receivers of volatility spillovers.

#### 5.2.2. Temporal Spillover Analysis

Static spillover measures the average spillover effects of crude oil and agricultural commodities markets during the sample period. This index, however, is not capable of capturing the time-varying property in the spillover effect. Empirically, market uncertainties usually affect the dependency structure across the markets over time. In terms of specification, volatility spillover will change during periods when volatilities change dramatically (e.g., financial crisis, European debt crisis, etc.); thus, it is necessary to select time spans incorporated with such events to dynamically measure the spillover.

This paper assumes that the rolling window equals 250 and the spillover duration equals 10. The volatility spillovers varying within the range of 5–63% are plotted in Figure 3, during which four representative economic periods can be identified. Period 1: 2005–2006. Hurricanes landfall in the Gulf of Mexico resulted in the shutdown of most crude oil production as well as a rapid decline in refining capacity. Oil prices briefly spiked from USD 36.2 to 79.15 per barrel and led to a sharp increase in volatility spillover, reaching the highest point of 63%. Period 2: 2007–2009. Return spillovers during this time period fluctuated between 8% and 20%. The global financial crisis triggered by the bankruptcy of the Lehman Brothers caused the volatility spillovers to skyrocket and then plummet drastically. In the meantime, crude oil also experienced price booms (USD 52.67 to 146.12) and subsequent busts (USD 146.12 to 40.06). Period 3: 2011–2018. Volatility spillovers swung between 10% and 30%. The European debt crisis spread to the entire eurozone in 2011 and provoked a rise in volatility spillovers. Thereafter, oil prices collapsed as the Organization of Petroleum Exporting Countries (OPEC) provided the market with excess supply in early 2015, while a reduction agreement was subsequently reached in the following year. Period 4: 2019—Present. COVID-19 has brought unexpected shocks to the global economy since the end of 2019. As the adverse impacts spread to all sectors, oil prices dropped to a low of USD 18.69 per barrel. Volatility spillovers stayed at an elevated level during this special stage until a new uptrend emerged in 2022. Two possible explanations for such radical changes refer to the Russian–Ukrainian war and the carbon neutrality target. On the one hand, risk aversion among investors has risen in line with the continuation of the Russian–Ukrainian war. As a major energy exporter, Russia’s crude oil export has been greatly restricted, due to which oil, as well as overall consumer prices, have been pushed up to record. On the other hand, all industrial countries are reducing their dependence on crude oil and promoting the development of green energy (e.g., solar energy, nuclear energy, tidal wave energy, etc.). In the international context of carbon neutrality, OPEC is cautious about building new production lines, and crude oil prices will be kept at a relatively high level as long as the current production is maintained. To conclude, the periodicity of total volatility spillovers is closely linked to the occurrence of global financial and economic events, indicating that the turmoil of the latter will exacerbate the spillover effects between crude oil and agricultural futures markets.

#### 5.2.3. Net Spillovers

Although the dynamic spillover provides information on the overall connectivity structure, it is not available to identify whether a variable is a net transmitter or a net receiver. Practically, this paper estimates the net spillovers to reveal the direction of volatility spillovers between crude oil and agricultural futures.

The rolling net volatility spillover effects over time are shown in Figure 4, from which the asymmetric and bidirectional properties can be detected by different responses to shocks. The net spillovers of WTI-σ fluctuated mildly between −2.5 and 2.5 in the early stage. The positive net spillovers appeared in 2013 when the European debt crisis happened and in 2017–2018 when developing economies rebounded steadily, which are two periods that observed a growth in the demand for crude oil. These results imply that the crude oil futures is a volatility transmitter, and a sudden economic event may lead to an increase in volatility spillover effects. The negative net spillovers are around 2015 and after 2020. In the year 2015, the price of WTI crude oil futures plummeted by 8% with the announcement of a cut-down on production made by OPEC. Late in 2021, major industrial economies, including China, the United States, Europe, Japan, and South Korea, basically reached a consensus on climate target plans to peak carbon emissions by 2030 and achieve carbon neutrality by 2060. The negative net spillovers suggest that WTI-σ has turned into a net receiver, and a sudden drop in crude oil prices may bring about an increase in spillovers from agricultural futures markets to the crude oil futures market. Additionally, among the agricultural futures market, the agricultural futures index, cotton, and soybean meal are the main volatility transmitters, while wheat and crude oil are the main volatility receivers. One hypothesis is that agricultural commodities such as soybean meal—the raw materials for biofuels and bioproducts—can be considered substitutes for crude oil to a certain extent.

## 6. Conclusions

This paper examines the time-varying Granger causality and the spillover effects between WTI crude oil futures and four sequences related to China’s agricultural futures, including the agricultural futures index, wheat hard futures, cotton futures, and soybean meal futures. Empirical evidence shows that crude oil futures are the time-varying Granger-cause of China’s agricultural futures during turbulent times such as financial crises, wars, and natural disasters. Moreover, the dynamics of volatility spillovers reveal the direction and degree of transmission during financial crises and economic turbulence over time.

Our empirical results are as follows. First, the linear Granger causality test results indicate that the Granger causality between international crude oil and soybean meal futures is bidirectional, whilst the others are unidirectional. In comparison, the time-varying Granger causality test shows significant results only when encountering special situations, such as major economic events and extreme natural disasters, and is also supported by a robust test under heteroskedasticity conditions. Second, the existence of bidirectional volatility spillovers in crude oil and agricultural futures is verified by the results of the DY spillover index. Such spillovers were exacerbated when the market or the international economic environment was undergoing a dramatic change.

The results of this research are expected to provide useful suggestions for many economic agents, such as international investors, speculators, and policymakers. With a comprehensive understanding of dynamic spillovers, investors can establish more effective risk-hedging models for the commodity futures markets, while policymakers can formulate appropriate policies to deal with financial risks and improve early warning capabilities. As mentioned above, the time-varying Granger causality test and dynamic spillover effects are highly dependent on the selected sample interval. An examination of the connectedness network and risk spillovers between crude oil and Chinese agricultural futures under short and long terms awaits future research.

## Figures and Tables

**Figure 1 ijerph-19-10593-f001:**
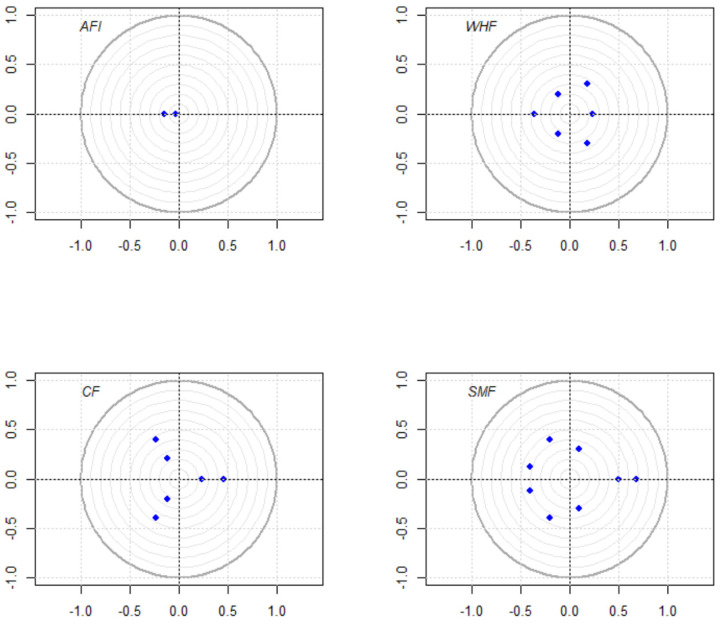
Stability test for VAR models.

**Figure 2 ijerph-19-10593-f002:**
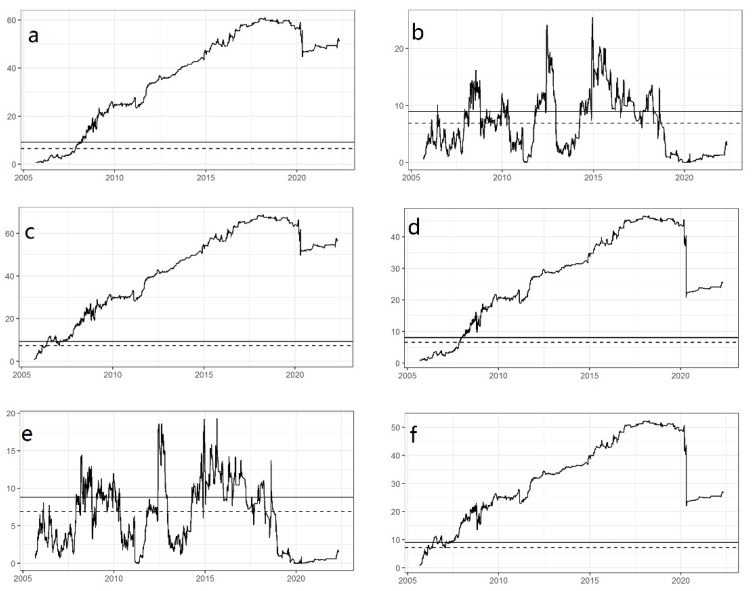
Results of time-varying Granger causality test (WTI→AFI). Note: The 90% and 95% confidence intervals are displayed with dotted and solid lines, respectively.

**Figure 3 ijerph-19-10593-f003:**
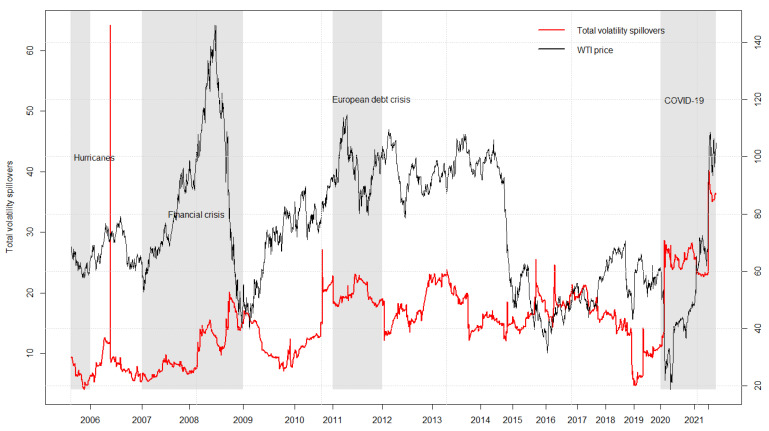
Total volatility spillovers and crude oil prices. Note: The shaded gray areas describe four specific intervals when major financial events happened during the sample period.

**Figure 4 ijerph-19-10593-f004:**
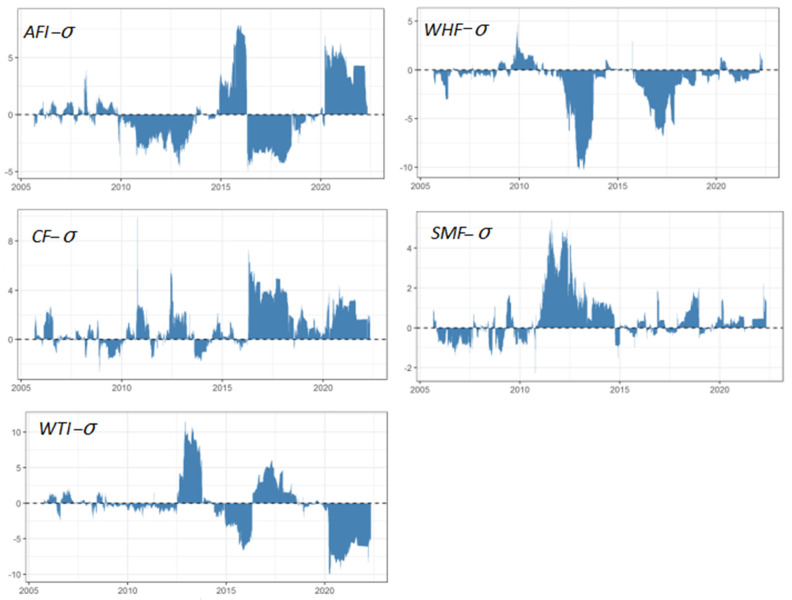
Net volatility spillovers.

**Table 1 ijerph-19-10593-t001:** Descriptive statistics.

Variable	Mean	Std.Dev.	Min	Max	Skewness	Kurtosis	ADF	JB
*AFI*	−0.001	0.781	−4.582	4.172	−0.175	5.799	−57.742 ***	0.000
*WHF*	0.042	1.608	−19.671	25.271	2.958	65.267	−54.172 ***	0.000
*CF*	0.008	1.050	−9.228	8.376	0.000	10.957	−50.763 ***	0.000
*SMF*	−0.056	2.120	−20.887	10.362	−1.744	17.650	−37.780 ***	0.000
*WTI*	0.004	2.506	−34.082	24.001	−1.210	28.869	−57.371 ***	0.000
AFI-σ	0.746	0.245	0.404	2.217	1.807	8.035	−4.737 ***	0.000
WHF-σ	2.095	0.361	1.607	3.381	1.065	3.301	−3.247 *	0.000
CF-σ	1.087	0.513	0.600	5.373	2.431	11.730	−12.600 ***	0.000
SMF-σ	1.883	1.268	1.111	17.042	4.289	31.761	−28.094 ***	0.000
WTI-σ	2.185	1.173	1.076	14.102	4.493	32.382	−6.159 ***	0.000

Note: * and *** denote the rejection of the null hypotheses of ADF test at the 10%, and 1% significance levels, respectively. JB provides the *p*-values from Jarque–Bera normality test.

**Table 2 ijerph-19-10593-t002:** Lag order test for WTI and AFI.

Lag	LL	LR	df	*p*	FPE	AIC	HQIC	SBIC
0	−9867.17				3.7478	6.9969	6.9985	7.0011
1	−9816.69	100.96 *	4	0	3.6263	6.9640	6.96854 *	6.97662 *
2	−9812.21	8.9568	4	0.062	3.6251 *	6.96364 *	6.9712	6.9847
3	−9809.3	5.8101	4	0.214	3.6279	6.9644	6.9751	6.9939
4	−9805.57	7.4616	4	0.113	3.6286	6.9646	6.9783	7.0025

Note: * represents the optimal lag order conducted by the corresponding test method.

**Table 3 ijerph-19-10593-t003:** Lag order test for WTI and WHF.

Lag	LL	LR	df	*p*	FPE	AIC	HQIC	SBIC
0	−11,937.8				16.2679	8.4650	8.4665	8.46916 *
1	−11,928.1	19.417	4	0.001	16.2022	8.4609	8.46546 *	8.4735
2	−11,923.4	9.3592	4	0.053	16.1944	8.4604	8.4680	8.4815
3	−11,912.2	22.391 *	4	0.000	16.112 *	8.45532 *	8.4660	8.4848
4	−11,909.4	5.7009	4	0.223	16.1251	8.4561	8.4698	8.4941

Note: * represents the optimal lag order conducted by the corresponding test method.

**Table 4 ijerph-19-10593-t004:** Lag order test for WTI and CF.

Lag	LL	LR	df	*p*	FPE	AIC	HQIC	SBIC
0	−10,721.2				6.8667	7.6024	7.6040	7.6067
1	−10,689.3	63.88	4	0.000	6.7320	7.5826	7.5872	7.59528 *
2	−10,683.9	10.836	4	0.028	6.7253	7.5816	7.5892	7.6027
3	−10,667.5	32.845 *	4	0.000	6.6663 *	7.57282 *	7.58346 *	7.6023
4	−10,664.1	6.6854	4	0.153	6.6694	7.5733	7.5870	7.6112

Note: * represents the optimal lag order conducted by the corresponding test method.

**Table 5 ijerph-19-10593-t005:** Lag order test for WTI and SMF.

Lag	LL	LR	df	*p*	FPE	AIC	HQIC	SBIC
0	−12,711.7				28.1580	9.0136	9.0151	9.0178
1	−12,538.7	345.96	4	0.000	24.9789	8.8938	8.8984	8.9064
2	−12,483.9	109.64	4	0.000	24.0949	8.8578	8.8654	8.8788 *
3	−12,473.1	21.545 *	4	0.000	23.9795	8.8530	8.8636 *	8.8825
4	−12,468.4	9.3239	4	0.053	23.9682 *	8.85248 *	8.8662	8.8904

Note: * represents the optimal lag order conducted by the corresponding test method.

**Table 6 ijerph-19-10593-t006:** Results of Granger causality test.

H_0_	*WTI* does not Granger-cause *AFI*		*AFI* does not Granger-cause *WTI*	
Lag	Chi-Sq.Statistic	Prob	Chi-Sq.Statistic	Prob
1	50.559	0.000	1.413	0.235
H_0_	*WTI* does not Granger-cause *WHF*		*WHF* does not Granger-cause *WTI*	
Lag	Chi-Sq.Statistic	Prob	Chi-Sq.Statistic	Prob
3	14.397	0.002	1.264	0.738
H_0_	*WTI* does not Granger-cause *CF*		*CF* does not Granger-cause *WTI*	
Lag	Chi-Sq.Statistic	Prob	Chi-Sq.Statistic	Prob
3	42.207	0.000	0.902	0.825
H_0_	*WTI* does not Granger-cause *SMF*		*SMF* does not Granger-cause *WTI*	
Lag	Chi-Sq.Statistic	Prob	Chi-Sq.Statistic	Prob
4	9.782	0.044	10.767	0.029

**Table 7 ijerph-19-10593-t007:** Wald test results for time-varying Granger causality.

Direction of Causality	Max Wald FE	Max Wald RO	Max Wald RE
$WTI\overset{GC?}{\to }AFI\;\left(Ι\right)$	60.996	25.44	68.812
	(6.560)	(6.850)	(7.327)
	[9.163]	[8.890]	[9.256]
$WTI\overset{GC?}{\to }AFI\;\left(ΙΙ\right)$	46.716	19.305	52.418
	(6.543)	(6.923)	(7.206)
	[8.109]	[8.805]	[9.016]
$WTI\overset{GC?}{\to }WHF\;\left(Ι\right)$	23.873	25.543	45.032
	(12.389)	(11.749)	(12.393)
	[14.503]	[15.799]	[15.799]
$WTI\overset{GC?}{\to }WHF\;\left(ΙΙ\right)$	15.355	12.844	21.799
	(12.852)	(12.726)	(12.948)
	[15.784]	[15.288]	[15.784]
$WTI\overset{GC?}{\to }CF\;\left(Ι\right)$	44.097	49.011	60.895
	(5.824)	(6.443)	(7.022)
	[7.930]	[8.815]	[8.835]
$WTI\overset{GC?}{\to }CF\;\left(ΙΙ\right)$	21.607	37.744	38.143
	(5.747)	(6.824)	(7.043)
	[8.016]	[8.333]	[8.802]
$WTI\overset{GC?}{\to }SMF\;\left(Ι\right)$	11.607	13.402	22.749
	(8.351)	(8.301)	(8.351)
	[11.642]	[11.653]	[11.653]
$WTI\overset{GC?}{\to }SMF\;\left(ΙΙ\right)$	10.711	23.804	24.024
	(6.698)	(7.217)	(7.647)
	[9.694]	[9.477]	[9.853]

Note: $x\overset{GC?}{\to }y$ shows whether variable x is the Granger cause of variable y. (Ι) and (ΙΙ) stand for homoskedasticity and heteroskedasticity. Values below in parentheses and square brackets are the 90% and 95% percentiles of the Wald statistics, respectively.

**Table 8 ijerph-19-10593-t008:** Static volatility spillover between crude oil and agricultural futures.

	WTI-σ	AFI-σ	WHF-σ	CF-σ	SMF-σ	From
WTI-σ	97.00	1.81	0.09	1.03	0.07	3.00
AFI-σ	0.66	86.25	0.49	6.39	6.21	13.75
WHF-σ	0.60	0.19	96.15	0.37	2.69	3.85
CF-σ	1.46	5.63	0.23	91.69	0.99	8.31
SMF-σ	0.08	5.35	2.11	1.41	91.05	8.95
To	2.80	12.99	2.91	9.20	9.96	7.57
NS	−0.20	−0.76	−0.95	0.89	1.02	

## Data Availability

All sample data sets are downloaded from the website. Data are available at http://www.nymex.com (accessed on 1 June 2022), http://www.51ifind.com and http://www.resset.com (accessed on 1 June 2022).

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
