# Peer review of "The Analysis of Causality and Risk Spillover between Crude Oil and China’s Agricultural Futures"

_ijerph, 2022, doi:10.3390/ijerph191710593_

Round 1

Author Response

Dear Reviewer,
We appreciate your valuable comments to our manuscript entitled “The Analysis of Causality and Risk Spillover Between Crude Oil and China’s Agricultural Futures” (ijerph-1876092). We have revised the manuscript based on your advice. Please refer to the attachment for specific modifications.

Reviewer 2 Report

The authors structurar way. Only several problem as follows: 

1.      The authors consider one index and three specific agricultural futures varieties as research objects. Regarding the latter, in order to show the representativeness of the selected futures, it is better to provide some necessary information such as the average transaction volume, the ranking or proportion of transaction size in the overall agricultural futures market.

2.      The window scale and spillover duration are assumed to be 250 and 10 in Section 5. 3.        The choice of such concrete numbers, however, is not discussed sufficiently. Therefore, the authors should explain why they make such assumptions for empirical analysis.

3.      There are several repeated statements contained in the article, in particular for those about concepts should be appropriately abridged. Here just to name a few examples for reference: concrete definitions or explanations on spillover effect, net spillover and DY index. The authors should carefully check the context and keep the article concisely overall.

4.      In Figure 3, the authors plot exclusively the volatility spillovers and identify four economic periods correspondingly, yet the following topic revolves around the relationship between spillovers and oil prices. This figure needs to be improved by presenting the international crude oil price changes as well.

5.      Equation (14): Notice the wrong order of i, j in numerator and i, j should be removed in the denominator.

Author Response

(The authors gave the same response as above.)
